# Red-TE Homozygous Alleles of *MdMYB10* Confer Full-Red Apple Fruit Skin in a High-Temperature Region

**Meili Wang** [1], **Yarong Wang** [1], **Tiyu Ding** [1], **Zhenli Yan** [1,2], **Zhe Zhou** [1,2], **Cuiying Li** [3], **Jia-Long Yao** [1,4,*] and **Hengtao Zhang** [1,2,*]

1   Zhengzhou Fruit Research Institute, Chinese Academy of Agricultural Sciences, 28 Gangwan Road, Zhengzhou 450009, China
2   Henan Key Laboratory of Fruit and Cucurbit Biology, Zhengzhou 450009, China
3   College of Horticulture, Northwest A&F University, Yangling, Xianyang 712100, China
4   The New Zealand Institute for Plant and Food Research Limited, Private Bag 92169, Auckland 1142, New Zealand
*   Correspondence: jia-long.yao@plantandfood.co.nz (J.-L.Y.); zhanghengtao@caas.cn (H.Z.)

**Abstract:** Apple is a major fruit crop grown worldwide and provides humans with an essential diet and health benefits. One of the health benefits is related to the accumulation of fruit anthocyanin, which also provides fruit with an attractive red colour. It is known that an *MdMYB10* allele containing a transposable element (TE) insertion in its promoter (termed Red-TE allele) underlies anthocyanin accumulation and red colouration in the fruit skin of cultivated apples. However, the distribution of this Red-TE allele in wider *Malus* germplasm accessions is not clear. In this study, we showed that *MdMYB10* RNA in fruit skin was specifically expressed from the Red-TE allele by using allele-specific expression analysis of transcriptome data. Apple cultivars and hybrids with homozygous Red-TE alleles showed stronger red colour than those with heterozygous alleles after analysing 65 cultivars and 337 hybrids. Furthermore, both hetero- and homozygous plants growing in the same high-temperature conditions had different colourations. However, the Red-TE allele was not detected in 16 wild apple accessions showing red skin, indicating that the red skin colour of these wild apples was not conferred by the Red-TE allele. These findings provide guidance for selecting cultivars able to develop consistent red colouration under high growth temperature conditions and open the opportunity for identifying novel genetic variants underpinning fruit red colouration in wild apple species.

**Keywords:** anthocyanin; allele-specific expression; *Malus*; MYB10; transcription factor; transposable element

## 1. Introduction

Apple fruit skin colour is a quality trait that affects fruit value and consumer preference and is selected in breeding programmes. Red colour, among all fruit skin colours, has been the most preferred colour by consumers. More and more red apple cultivars have been bred, and many new cultivars have fully red fruit skin [1]. The development of the red apple skin colour is associated with the accumulation of anthocyanin. In addition, to improving fruit's visual appearance, anthocyanin also provides anti-oxidation and anti-arteriosclerosis effects that are believed to reduce the risk of cancer and heart disease [2].

Anthocyanin biosynthesis is regulated by a transcription factor complex, including MYB10, bHLH and WD40 [3,4]. In apples, MdMYB10 (also known as MdMYB1) is the core regulator for anthocyanin biosynthesis in fruit skin [3,5]. The expression of *MdMYB10* is regulated by temperature, light, and DNA methylation. Hot air treatments can rapidly reduce *MdMYB10* transcription in apple fruit [6], while cold treatments can increase the expression of *MdbHLH3*, *MdMYB10* and anthocyanin biosynthesis pathway genes [7]. Light promotes anthocyanin synthesis by promoting the stability of MdMYB1 protein in apple skin [8]. The methylation level of the *MdMYB1* promoter in apple skin also affects the red

colouration of the skin. Bagged fruit can develop redder skins after the bag is removed than unbagged fruit because the bagging treatment reduces the methylation level in *MMYB1* promoter regions [9].

Apples grown in lower-temperature regions tend to have a high level of redness in fruit skin; however, the same is not true in high-temperature regions, thereby reducing the value of apples in these regions. Breeding programmes are now aimed at selecting red apple cultivars for high-temperature regions. An effective genetic selection marker would facilitate such breeding.

Three genetic variants in the promoter region that regulate *MdMYB10* expression have been identified. First, a minisatellite named R6 repeat provides binding sites for MdMYB10 to auto-regulate its own expression and confers the red colour of leaves, flowers, and fruit flesh and skin [10]. Second, a long-terminal-repeat (LTR) type transposable element (TE) named Red-TE enhances *MdMYB10* expression and anthocyanin biosynthesis in fruit skin [11]. Third, terminal-repeat retrotransposons in miniature (TRIM) type TE promote *MdMYB10* expression and anthocyanin biosynthesis in flower petals [12].

Of these variants, the Red-TE causes fruit's specific skin colouration and is chosen for the development of red fruit skin markers. The previous study has shown that the 112 red cultivars tested all contained the Red-TE, but none of the 33 non-red cultivars tested contained the Red-TE [11]. However, it is not known the allele composition in these cultivars or whether homozygous Red-TE alleles provide stronger skin redness than heterozygous alleles. Of 14 wild apple accessions tested, three contained the Red-TE alleles [11]. It is unclear if the Red-TE is also a major regulator of the red fruit skin of small wild apples.

This study aimed to answer the questions of whether the Red-TE allele composition affects the strength of fruit skin redness and whether it is responsible for the red skin of small wild apples. Our results indicate that homozygous Red-TE alleles confer stronger red colour than the heterozygous alleles, in particular, in an orchard with high summer temperature. Furthermore, the red skin colour of wild apple species with small fruit is not conferred by Red-TE. These findings are important for guiding apple breeding programmes to select cultivars with consistent red colouration under different growth temperatures and open the opportunity for identifying new genetic variants regulating apple fruit red colouration using accessions of wild apple species.

## 2. Materials and Methods

### 2.1. Plant Materials

Leaf and fruit samples of 65 cultivated varieties and 337 hybrids of 'NY543' × 'Pink Lady' were collected from the experimental apple orchard at the Zhengzhou Fruit Research Institute CAAS. Leaf and fruit samples of 25 wild species were collected from the experimental orchard at Zhengzhou Fruit Research Institute CAAS and Northwest A&F University.

Samples were frozen in liquid nitrogen immediately after being collected into a 2 mL centrifuge tube and then stored at −80 °C. From June to October, 5–15 whole fruit samples of each plant were randomly collected and taken to the laboratory for fruit skin colour phenotyping.

### 2.2. Genome and Transcriptome Sequencing and Analyses

For genome sequencing, the protocol used was similar to a previously described method [12]. In brief, DNA extracted from leaves was used to construct sequence libraries with an insert size of 400 bp. The liberties were sequenced on the HiSeq2000 platform. The sequence reads were aligned to the apple reference genome GDDH13 [13] using bowtie (version 2.2.5) to generate BAM files that were visualised using Integrated Genome Viewer (IGV).

For transcriptome sequencing, total RNA was extracted from fruit skin tissues of mature 'Royal Gala' fruit and used for sequence libraries construction and Illumina sequencing based on the method described in a previous study [14]. The sequence reads were cleared and aligned

to the apple reference genome GDDH13 [13] using the previously described protocols [14] to generate BAM files that were visualised using Integrated Genome Viewer (IGV).

### 2.3. DNA Extraction and PCR Analysis

Genomic DNA was extracted from young leaf tissues using a DNA extraction kit (Plant genome DNA extraction kit, Luoyang Aisen Biological Technology Co. Ltd.Luoyang, China) according to the manufacturer's instructions and used as templates for PCR analyses to genotype Red-TE insertion in *MdMYB10* promotor. The PCR primers were designed based on *MdMYB10* promoter sequences in the GDDH13 (ID: MD09G1278600) and HFTH1 (HF36879-RA) reference genomes [11,13]. The $F_1$ primer sequence (CGGATTGTTCCTGCT-GTCTCTCTGTTGACA) is located in the Red-TE of the HF reference genome, but the F2 primer sequence (ATATCACACTCCCTTCTCTTTCTAG) is not in the Red-TE. The F1 and R1 primer (TTTTCCCTTCATTGAGCACTAATTTTC) can amplify a 386 bp DNA fragment for the Red-TE allele, while the F2 and R1 primers can amplify a 250 bp DNA fragment for the allele without the Red-TE. The PCR was carried out using a pre-denaturation step at 94 °C for 2.5 min, then 33 cycles of 94 °C for 1 min, 55 °C for 1 min and 72 °C for 1.5 min, and a final extension step at 72 °C for 10 min. The PCR products were analysed by electrophoresis in 2% agarose gel containing ethidium bromide and photographed under a UV light.

### 2.4. Measuring Apple Fruit Skin Color

Five fruits were collected randomly from a single tree of each cultivar or hybrid plant grown in Zhengzhou (E 112°42′–114°13′, N 34°16′–34°58′, 108 m), with high temperatures in summer and autumn at the fruit maturity stage. 10–15 fruits were collected from each wild apple germplasm accession grown in Zhengzhou and Xi'an.

The apple skin colour was determined by using the Chromameter CR-400 (Minolta, Japan) and the CLEAB colour space method. [15] For each plant, 5–15 fruit were analysed, and each fruit was analysed at 5 different points on the fruit skin. The skin colour was defined by the three-dimensional coordinate space of brightness value (L), red/green (±a) and yellow/blue (±b). L (range 0–100) shows the light, and shade degree, a (scope ± 60) shows red and green values, positive values indicate red, and negative values indicate green. b (range ± 60) shows blue and yellow values; positive is yellow, and negative is blue. The higher the absolute values of L, a, and b, the darker the skin colour.

## 3. Results

### 3.1. Allele-Specific Expression of MdMYB10 in Apple Fruit Skin

To determine the allele-specific expression and its association with Red-TE insertion, DNA and fruit skin RNA sequence reads of apple cultivars were aligned to the GDDH13 reference genome. The alignments were viewed using IGV. The alignment of RNA sequence reads of apple cultivar NY543 to the reference genome showed a new intron in the 5′ untranslated region (5′-UTR) of *MdMYB10*, which is not annotated in the reference genome (Figure 1a). The alignment of DNA sequence reads of NY543 showed a DNA insertion in the promoter and a homozygous SNP in intron 3 of *MdNYB10* of NY543 (Figure 1a). The insertion site is the same as the Red-TE insertion site. Further analysis showed this insertion being homozygous in NY543, heterozygous in 'Royal Gala' and absent in 'Granny Smith'. The insertion caused a short sequence (ATATG) duplication (Figure 1b). A heterozygous SNP was also identified in intron 3 of *MdMYB10* of 'Royal Gala' according to the DNA sequence reads, but the SNP was homozygous according to the RNA sequence reads, indicating only one allele expressed, which is the allele that is different from the reference genome and contains the Red-TE (Figure 1b,c). The detection of RNA sequence reads in the intron region indicates alternative splicing.

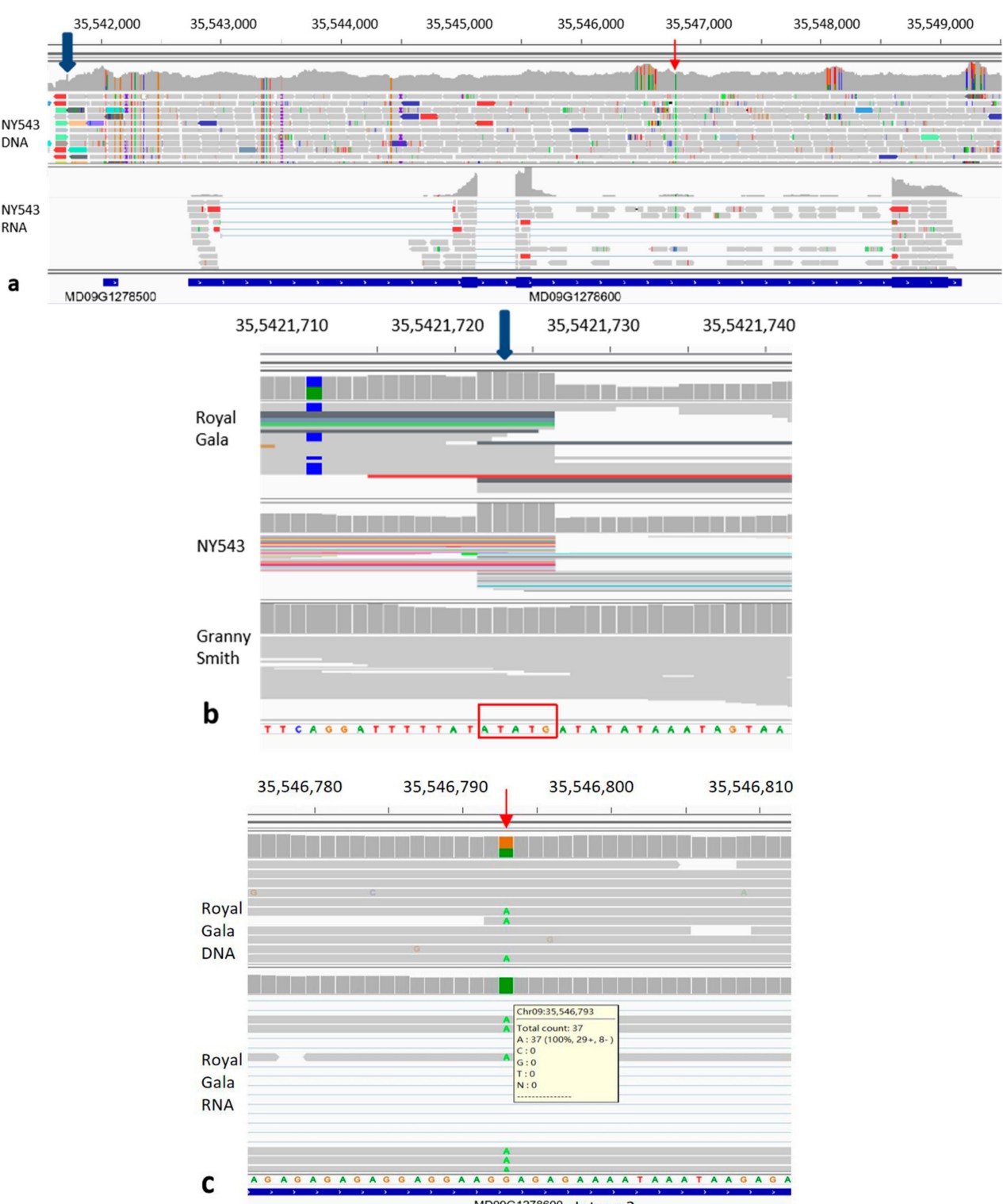

**Figure 1.** *Cont.*

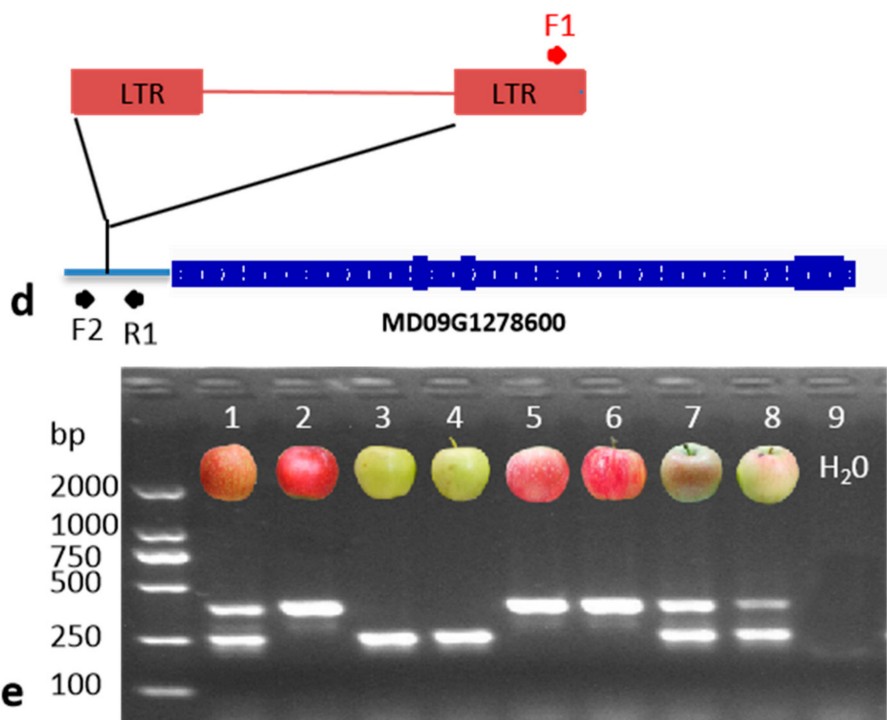

**Figure 1.** Genome and transcriptome analyses of *MdMYB10* locus. (**a**), DNA and fruit skin RNA sequence reads of the apple line NY543 were mapped to the GDDH13 reference genome to generate bam files that were viewed using IGV. The blue arrow indicates the site of Red-TE insertion, and the red arrow indicates an SNP site in intron 3. (**b**), An IGV view covering the Red-TE insertion site shows the duplicated sequence due to TE insertion (marked with a red square), heterozygous insertion in the genome of 'Royal Gala', homozygous insertion in NY543 and no insertion in 'Granny Smith'. How this IGV view can show the TE insertion site is explained in Figure S1. (**c**), An IGV view of the SNP in intron 3 shows the allele-specific expression of RNA of SNP$^A$, which was from the allele containing the Red-TE. (**d**), A diagram shows the location of three PCR primers, F1 in the long terminal repeat (LTR) of the Red-TE, F2 and R1 flanking the Red-TE insertion site. (**e**), PCR DNA fragments were amplified from genomic DNA samples extracted from eight apple cultivars (1–8, 'Pink Lady', NY543, 'Jinqing', 'Yuguan', 'Idared', 'Bella', 'Ace', 'Rome Beauty') using the three primers of (**c**). The two amplified DNA fragments, 386 bp and 250 bp created using F1/R1 and F2/R1 primers, represent the two *MYB10* alleles with or without the Red-TE insertion, respectively.

### 3.2. The Distribution of MdMYB10_Red-TE Allele in Apple Cultivars

Using the primers F1/F2/R1 (Figure 1d), two PCR DNA fragments, 250 and 386 bp, were amplified from apple cultivars (Figure 1e). The 250 bp fragment was created using F2/R1 primers, representing the *MYB10* allele without the Red-TE insertion (named as rt), while the 386 bp fragment was created using F1/R1 primers, representing the *MYB10* allele with the Red-TE insertion (named as RT). The 386 bp fragment was detected from three cultivars, NY543, 'Bella', and 'Idared', indicating that they were Red-TE homozygous (RTRT). The three cultivars also showed strong red skin. Both 386 and 250 bp fragments were detected from the other three cultivars ('Pink Lady', 'Ace', and 'Rome Beauty'), indicating that they were Red-TE heterozygous (RTrt). The heterozygous cultivars showed weak and patched red colour on the fruit skin. Only the 250 bp fragment was detected in two cultivars ('Jinqing' and 'Yuguan'), showing no red skin, indicating that the two non-red cultivars were homozygous for non-Red-TE insertion (rtrt; Figure 1e). The correlation between the presence of Red-TE and *MdMYB10* expression is shown in Figure 1c and in a previous publication [11]. This result suggested that homozygous Red-TE alleles were stronger than heterozygous alleles in conferring red skin colour.

Follow on that, the fruit skin colours of 65 apple cultivars were analysed with a chromameter. *MdMYB10* allele-types of them were analysed by using PCR. The analyses divided the cultivars into four groups. Group one contained 11 cultivars with yellow or green skin and without Red-TE insertion (rtrt). Group two contained 29 cultivars with red skin and heterozygous Red-TE insertion (RTrt). Group three contained 17 cultivars with red skin and homozygous Red-TE (RTRT) (Table 1). The skin colours of these three groups matched the Red-TE genotype as expected, i.e., cultivars with yellow or green skin contained no Red-TE insertion, while cultivars with red skin contained at least one copy of Red-TE insertion. However, the eight cultivars in group four did not show the expected match between skin colour and the Red-TE genotype. Five red cultivars ('Hongjin', 'Kuihua', 'Gloster', 'Xinlimei', and 'Summer Champion') contained no Red-TE insertion allele, indicating that another mechanism may regulate fruit skin colour. Three yellow cultivars ('Iowa Beauty'', 'Tianyu', and 'Grimes Golden') contained one copy of the Red-TE insertion allele (Table 1). This inconsistency between fruit skin colour and the Red-TE genotype may be caused by climate conditions leading to the skin colour variation, i.e., cultivars with heterozygous Red-TE may not be able to develop red skin in high-temperature regions. After the skin colour was analysed by using a chromameter, the average a/b value (0.942) of 17 red cultivars with RTRT genotype was significantly ($p > 0.05$ by *t*-test) higher than that (0.632) of 29 cultivars with RTrt genotype (Table 1), indicating that RTRT confers a higher level of redness of fruit skin in Zhengzhou region with high temperature.

**Table 1.** Analysis of fruit skin colour and *MdMYB10* genotype in 65 apple cultivars.

| Cultivar Name | Fruit Skin Colour | MdMYB10 Genotype | a/b Ratio | |
|---|---|---|---|---|
| | | | **Mean** | **SD** |
| Jingqing | Yellow | rtrt | −0.114 | 0.424 |
| Ozark Gold | Yellow | rtrt | −0.210 | 0.111 |
| Gold Spur | Yellow | rtrt | −0.228 | 0.042 |
| Cox's Orange Pippin 363 | Yellow | rtrt | −0.263 | 0.045 |
| Beidou | Yellow | rtrt | −0.308 | 0.085 |
| Winter Banana | Yellow | rtrt | −0.341 | 0.063 |
| Yuguan | Yellow | rtrt | −0.422 | 0.051 |
| Jingyuan | Yellow | rtrt | −0.427 | 0.006 |
| Mosiketouming | Yellow | rtrt | −0.456 | 0.068 |
| Granny Smith | Green | rtrt | | |
| Indo | Green | rtrt | −0.600 | 0.029 |
| Regent | Red | RTrt | 1.296 | 0.500 |
| Royal Gala | Red | RTrt | | |
| Red 210 | Red | RTrt | 1.274 | 0.211 |
| Pink Lady | Red | RTrt | 1.102 | 0.114 |
| Summerland | Red | RTrt | 0.985 | 0.100 |
| Close | Red | RTrt | 0.968 | 0.395 |
| Fuli | Red | RTrt | 0.871 | 0.298 |
| De No.6 | Red | RTrt | 0.861 | 0.327 |
| Starking Delicious | Red | RTrt | 0.850 | 0.228 |
| Monroe | Red | RTrt | 0.785 | 0.135 |
| Huamei | Red | RTrt | 0.718 | 0.265 |
| Black Ben Davis | Red | RTrt | 0.709 | 0.168 |
| Fuji Spur | Red | RTrt | 0.695 | 0.123 |
| Yuejin | Red | RTrt | 0.680 | 0.259 |
| Starkrimson | Red | RTrt | 0.646 | 0.103 |
| Hardi Brite | Red | RTrt | 0.638 | 0.300 |
| Gongqiduanfu | Red | RTrt | 0.610 | 0.170 |
| Rome Beauty | Red | RTrt | 0.577 | 0.275 |
| Ningqiu | Red | RTrt | 0.556 | 0.417 |
| Senshu | Red | RTrt | 0.470 | 0.370 |
| Kogetsu | Red | RTrt | 0.468 | 0.069 |
| Braeburn | Red | RTrt | 0.453 | 0.230 |
| Fameuse | Red | RTrt | 0.394 | 0.370 |
| Red Jonagold | Red | RTrt | 0.300 | 0.130 |
| Ace | Red | RTrt | 0.215 | 0.210 |
| Anglin | Red | RTrt | 0.194 | 0.125 |
| Discovery | Red | RTrt | 0.172 | 0.003 |
| Yanfeng | Red | RTrt | 0.164 | 0.097 |
| Hongxi Jinqing | Red | RTrt | 0.049 | 0.002 |

**Table 1.** *Cont.*

| Cultivar Name | Fruit Skin Colour | MdMYB10 Genotype | a/b Ratio | |
|---|---|---|---|---|
| | | | Mean | SD |
| GS-58 | Red | RTRT | 1.854 | 0.177 |
| NY543 | Red | RTRT | 2.240 | 0.479 |
| Liberty | Red | RTRT | 1.815 | 0.679 |
| Mollie's | Red | RTRT | 1.423 | 0.726 |
| Ralls | Red | RTRT | 1.192 | 0.298 |
| Bella | Red | RTRT | 1.055 | 0.343 |
| Luxiang | Red | RTRT | 0.996 | 0.467 |
| Cox's Orange Pippin 312 | Red | RTRT | 0.980 | 0.311 |
| Meize | Red | RTRT | 0.948 | 0.111 |
| Hongbaoshi | Red | RTRT | 0.888 | 0.388 |
| Idared | Red | RTRT | 0.857 | 0.585 |
| Changfu 2 | Red | RTRT | 0.780 | 0.403 |
| Longjinmi | Red | RTRT | 0.754 | 0.195 |
| Huadan | Red | RTRT | 0.690 | 0.426 |
| Xinshijie | Red | RTRT | 0.480 | 0.131 |
| Huayu | Red | RTRT | 0.240 | 0.287 |
| Zhongqiu | Red | RTRT | 0.112 | 0.207 |
| Gloster | Red | rtrt | 1.224 | 0.276 |
| Summer Champion | Red | rtrt | 1.109 | 0.419 |
| Xinlimei | Red | rtrt | 0.880 | 0.553 |
| Hongjin | Red | rtrt | 0.805 | 0.458 |
| Kuihua | Red | rtrt | 0.570 | 0.097 |
| Iowa Beauty | Yellow | RTrt | 0.025 | 0.199 |
| Tianyu | Yellow | RTrt | −0.262 | 0.003 |
| Grimes Golden | Yellow | RTrt | −0.398 | 0.070 |

*3.3. Homozygous MYB10_Red-TE Allele Conferring Improved Red Fruit Skin Colour in High-Temperature Regions*

To confirm the above result that homozygous Red-TE can improve apple fruit skin colour, we further analysed 337 hybrids of a cross between NY543 and 'Pink Lady'. The hybrid trees were also grown in Zhengzhou, where the day temperature is high and the temperature difference between day and night is low in the summer and autumn, which is not ideal for fruit colouration. NY543 is Red-TE homozygous (Figure 1e) and showed full and bright red skin (Figure 2a). 'Pink Lady' is Red-TE heterozygous (Figure 1e) and showed patched and medium-level red skin (Figure 2b). The hybrids were divided into two types based on skin colour, with full bright red skin or patched red skin (Figure 2c,d). PCR analysis showed that the hybrids with full bright red skin were Red-TE homozygous (RTRT), while hybrids with patched and medium-level red skin were Red-TE heterozygous (RTrt) (Figure 2e). In total, there were 165 RTRT, 170 RTrt and 2 rtrt plants (Figure 2f). The two rtrt plants shouldn't be part of the controlled cross. The segregation ratio between RTRT and RTrt was 1:1 by the $\chi^2$ test. The skin colour of the hybrids was also analysed with a chromameter. The a/b value of the 170 RTRT hybrids was significantly higher (based on unpaired *t*-test, $p < 0.05$) than that of the 165 RTrt hybrids (Figure 2g, Table S1), indicating Red-TE homozygous alleles can further enhance apple skin colour development compared to the heterozygous alleles in a high-temperature region.

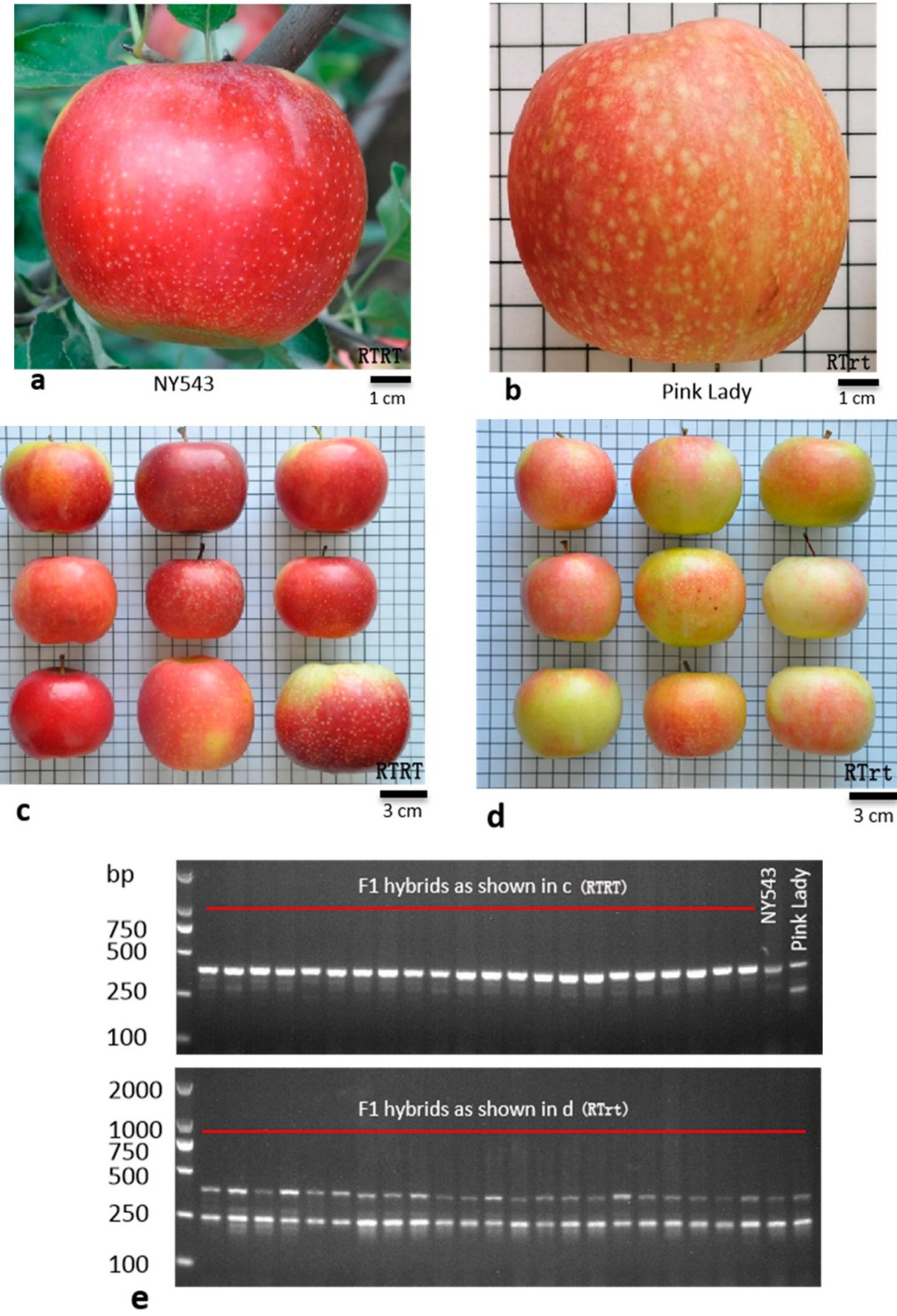

**Figure 2.** *Cont.*

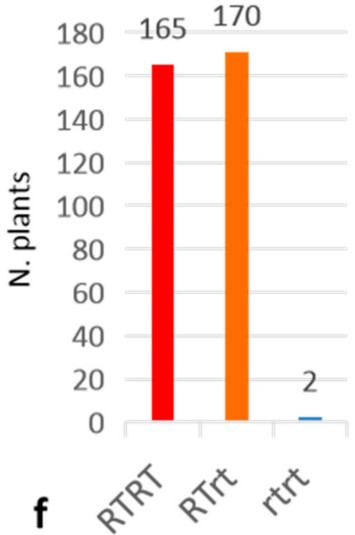
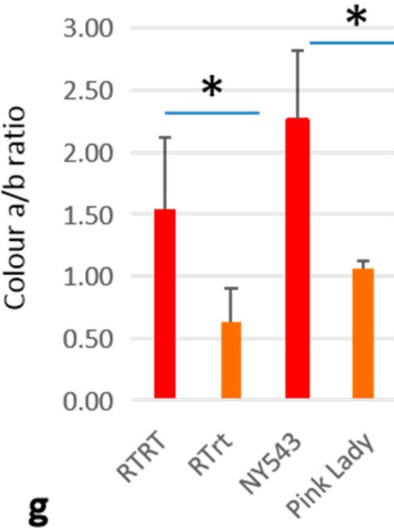

**Figure 2.** The dosage of the Red-TE allele is positively associated with fruit skin redness. (**a**–**d**), photographs show mature fruit of paternal parent NY543 (**a**), maternal parent 'Pink Lady' (**b**) and nine F1 hybrids with full red fruit skin (**c**) and other nine F1 hybrids with medium levels of fruit redness (**d**). (**e**), PCR analysis showed homozygous Red-TE insertion in NY543 and F1 hybrids, as shown in (**c**), and heterozygous Red-TE insertion in 'Pink Lady' and F1 hybrids, as shown in (**d**). (**f**), segregation analysis of two different *MYB10* alleles in 337 F1 hybrids showed 165 Red-TE homozygous (RTRT) and 170 heterozygous (RTrt) hybrids. (**g**), colour ratios (**a**,**b**) were measured using a chromameter. * presents a significant difference based on an unpaired *t*-test ($p < 0.05$).

### 3.4. MYB10_Red-TE Allele Is Not Responsible for Red Fruit Skin in Small Wild Apples

To determine whether the Red-TE insertion is also responsible for red fruit skin in wild apples, we genotyped 25 wild apple accessions with different skin colours. The skin colour ranged from green and yellow to different levels of redness, as shown in Figure 3a–h. Twenty-two accessions had no Red-TE insertion, and three (including Binzi, Shanxihaitanghua and Yingyehaitang) had Red-TE insertion (Figure 3i, Table 2). The three accessions with Red-TE insertion showed red fruit skin. Eighteen of the 22 accessions without Red-TE insertion also showed red skin, and the other four accessions showed green or yellow skin (Table 2). This result indicates the red skin in the majority of wild apples is not associated with the Red-TE insertion in the promoter region of the *MdMYB10* gene.

**Table 2.** Analysis of fruit skin colour and *MdMYB10* genotype in 25 wild apple accessions.

| Accession Name | Species Name | Fruit Skin Colour | *MdMYB10* Genotype | a/b Ratio | |
|---|---|---|---|---|---|
| | | | | Mean | SD |
| Lijiangshandingzi | *M. rockii* | Red | rtrt | 1.534 | 0.529 |
| Yingyehaitang-1 | *M. prunifolia* | Red | rtrt | 1.024 | 0.326 |
| Shanxihaitanghua(zz) | *M. robusta* | Red | RTRT | 2.192 | 0.135 |
| Hongsanyehaitang | *M. sieboldii* | Red | rtrt | 1.382 | 0.602 |
| Sanyehaitang | *M. sieboldii* | green | rtrt | −0.522 | 0.032 |
| Dongbeihuanghaitang | *M. honanensis* | Yellow | rtrt | −0.064 | 0.067 |
| Lushihongguo | *M. honanensis* | Green | rtrt | −0.629 | 0.039 |
| Baodehaihong(zz) | *M. micromalus* | Red | rtrt | 3.832 | 0.376 |
| Shandingzi | *M. baccata* | Red | rtrt | 2.830 | 0.690 |
| Maoshandingzi | *M. manshurica* | Red | rtrt | 2.680 | 0.455 |

Table 2. *Cont.*

| Accession Name | Species Name | Fruit Skin Colour | *MdMYB10* Genotype | a/b Ratio | |
|---|---|---|---|---|---|
| | | | | Mean | SD |
| Jinxibeishandingzi-1 | *M. baccata* | Red | rtrt | 2.466 | 0.627 |
| Jixibeishandingzi-2 | *M. baccata* | Red | rtrt | 2.197 | 0.497 |
| Wushanbianyehaitang | *M. toringoides* | Red | rtrt | 1.914 | 0.521 |
| Maoshandingzi-1(zz) | *M. manshurica* | Red | rtrt | 1.903 | 0.537 |
| Hongshijie(zz) | unknown | Red | rtrt | 1.864 | 0.080 |
| Hongxunyihao(zz) | *M. sieversii* | Red | rtrt | 1.627 | 0.276 |
| Yingyehaitang-2(zz) | *M. prunifolia* | Red | RTRT | 0.843 | 0.075 |
| Binzi | *M. asiatica* | Red | RTrt | 0.833 | 0.230 |
| Gansuhaitang | *M. kansuensis* | Red | rtrt | 0.761 | 0.271 |
| Xiaojinhaitang | *M. xiaojinensis* | Red | rtrt | 0.491 | 0.658 |
| Zhaai-76 | *M. baccata* | Red | rtrt | 0.415 | 0.231 |
| Maoshandingzi-2(zz) | *M. manshurica* | Orange | rtrt | 0.357 | 0.165 |
| Duohuahaitang | *M. floribunda* | Red | rtrt | −0.043 | 0.207 |
| pingyitiancha | *M. hupehensis* | Yellow | rtrt | −0.156 | 0.174 |
| Hubeihaitang | *M. hupehensis* | Yellow | rtrt | −0.338 | 0.287 |

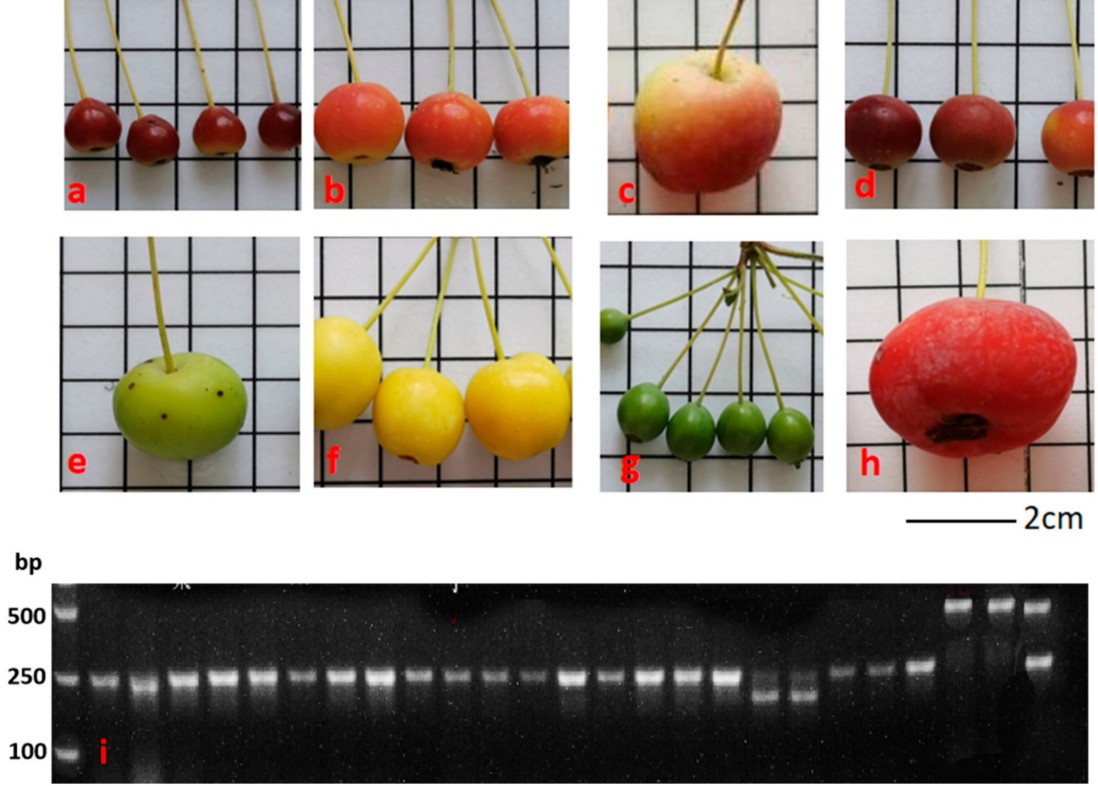

**Figure 3.** Wild apple fruit skin colour and *MYB10* allele compositions. (**a**–**h**) photographs show the mature fruit of eight wild apple accessions. (**a**) *M. rocki* Lijiangshandingzi, (**b**) *M prunifolia* Yingyehaitang, (**c**) *M robusta* Shangxihaitanghua, (**d**) *M sieboldii* Hongsanyehaitang, (**e**) *M sieboldii* Sanyehaitang, (**f**) *M honanensis* Dongbeihuanghaitang, (**g**) *M honanensis* Lushihongguo, (**h**) *M micromalus* Baodehaihong. These are the same as the first eight accessions listed in Table 2. (**i**) PCR analysis using the primers described in Figure 1c showed that Red-TE was absent in 22 accessions, homozygous in two and heterozygous in one accession of wild apple species.

## 4. Discussion

It is known that Red-TE insertion in the promoter of *MdMYB10* enhances *MdMYB10* expression in apple fruit skin, thus anthocyanin biosynthesis and red colouration [11]. However, the genotype of Red-TE insertion is not clear for most apple cultivars. The PCR primers used to detect the presence of Red-TE cannot distinguish homozygous from heterozygous Red-TE insertion [11]. In order to determine Red-TE genotypes in many apple cultivars, we used a mix of three primers that classify cultivars into three types, Red-TE homozygous (RTRT), Red-TE heterozygous (RTrt) and non-Red-TE insertion (rtrt). Based on the PCR analysis with these primers, we have shown that, in general, apple cultivars with red fruit skin contain at least one copy of the Red-TE insertion, and those with green or yellow skin contain no Red-TE insertion. The cultivars with two copies of the Red-TE can develop a higher level of redness than the cultivars with one copy of the Red-TE (Figure 1e, Table 1).

However, there were some unexpected results on the correlation between the presence of Red-TE and the colour of fruit skin in a number of cultivars. Five cultivars ('Hongjin', 'Kuihua', 'Gloster', 'Xinlimei' and 'Summer Champion') showed red fruit skin but contained no Red-TE, indicating that another mechanism may regulate fruit skin colour in these cultivars. It is known that in addition to the Red-TE, other two variants, a minisatellite repeat [10] and a TRIM TE [12] in *MdMYB10,* can also upregulate *MdMYB10* expression, thus anthocyanin accumulation and red colouration. It is most likely that novel variants enhancing *MdMYB10* expression are remained to be discovered.

Furthermore, three cultivars ('Iowa Beauty", 'Tianyu' and 'Grimes Golden') contained one copy of the Red-TE insertion but did not develop red fruit skin. This may be caused by the high-temperature conditions in the growing region. It is known that some cultivars cannot develop red colour in regions with high temperatures, even if they develop red colour in regions with relatively low temperatures [6]. An alternative explanation may be the presence of inhibition of anthocyanin biosynthesis and accumulation in these cultivars, such as methylation of *MdMYB10* promotor [16] or expression of repressor MYB proteins [17].

The presence of Red-TE in wild apple species is also not clear. Reference genome analysis showed that the Red-TE was present in *M. sieversii* but not linked to *MdMYB10*, while not present in the *M. sylvestris* genome [18]. This analysis could not show the frequency of the Red-TE presence in the wild apples as the reference genome was assembled using one accession of the species. PCR analysis of 14 wild apple accessions showed the presence of Red-TE in one accession of *M. sieversii, M. baccata* × unknown, and *M. robusta* × unknown [11]. However, this PCR could not unequivocally show whether the TE is linked to the *MdMYB10* gene because the pair of PCR primers used are both located within the TE, and no *MdMYB10*-specific PCR primers were used. In this study, we used an *MdMYB10*-specific PCR primer and a Red-TE primer to detect the specific presence of Red-TE in *MdMYB10.* Red-TE was detected in three wild apple accessions showing red fruit skin but not detected in other 16 red or six non-red wild accessions (Table 2). The low frequency of Red-TE presence in wild apples may be due to a genetic admixture caused by gene flow from cultivated to wild apples, which is noted in *Malus* [19,20]. The absence of Red-TE in 16 wild apple accessions suggests that other elements rather than Red-TE regulate the red fruit skin of wild apples.

Because of global warming, it is important to breed apple cultivars that can develop full red fruit in high-temperature regions. This study discovered that homozygous Red-TE could confer full red skin in high-temperature conditions. The discovery provides guidance for future breeding programmes to generate red apple cultivars suitable for high-temperature regions.

## 5. Conclusions

We first showed that the *MdMYB10* was specifically expressed from the Red-TE allele using allele-specific expression analysis with transcriptome data and then analysed the

Red-TE allele compositions in 65 apple cultivars, 337 hybrids of a breeding family and 25 wild apple accessions using PCR analysis. Our results showed that red-skinned cultivars and hybrids contained homozygous or heterozygous Red-TE alleles, and the homozygous plants showed stronger red colour than the heterozygous plants in the same high summer temperature conditions. However, the red skin colour of most wild apple accessions was unlikely conferred by the Red-TE. These findings are important for guiding apple breeding programmes to select cultivars with consistent red colouration under high growth temperatures and open the opportunity for identifying new genetic variations regulating apple fruit colouration using accessions of wild apple species.

**Supplementary Materials:** The following supporting information can be downloaded at: https://www.mdpi.com/article/10.3390/horticulturae9020270/s1, Figure S1: How to identify a TE insertion site using IGV. A short sequence in a genome (reference genome) may become a target sit for TE insertion (a). During TE insertion, the short target sequence is duplicated. The Next Generation Sequence (NGS) reads are generated from the TE insertion mutant allele (b). When the NGS reads are mapped to the reference genome, the reads that are marked within the two circles in (b) and contain mostly the TE sequences can not be mapped to the reference. The rest reads containing the target sequence are mapped. Due to the duplication of the target sequence, mapping coverage is higher at the target site than the adjacent sites (c). This high mapping coverage can be seen using IGV as an indication of TE insertion site. Table S1: hybrid.

**Author Contributions:** J.-L.Y. and H.Z. designed the study. M.W. and Y.W. performed experiments. T.D., Z.Y., Z.Z. and C.L. supplied experimental materials; J.-L.Y., M.W. and H.Z. wrote the manuscript. All authors have read and agreed to the published version of the manuscript.

**Funding:** This study was funded by The Agricultural Science and Technology Innovation Program (CAAS-ASTIP-2021-RIP-02), the Major Science and Technology Project of Henan Province (221100110400) and China Agriculture Research System (CARS-27).

**Data Availability Statement:** No new data were created in this study.

**Conflicts of Interest:** The authors declare no conflict of interest.

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
