# Peer review of "Red-TE Homozygous Alleles of MdMYB10 Confer Full-Red Apple Fruit Skin in a High-Temperature Region"

_horticulturae, doi:10.3390/horticulturae9020270_

Round 1

Reviewer 1 Report

The manuscript “Red-TE homozygous alleles of MdMYB10 confer full-red apple 2 fruit skin in a high temperature region” by Wang et al., presents a thorough study for identifying novel genetic variants that can help cultivators to select a better apple variety with red skin color and can proliferate well in higher temperature regions.

Overall, the study seems to be carefully conducted and the results are presented clearly and in a concise manner. Specific comments are mentioned below.

1.      Abstract- “Furthermore, the homozygous plants showed consistently strong red coloration in a growing region with high temperature where the heterozygous plants developed no or little red coloration” rephrase this sentence to show that both, hetero and homozygous plants growing in same high temperature conditions had different colorations.

2.      Line 52-53 “Many apple cultivars cannot develop high level of fruit skin redness in production regions with high temperature even if they can develop in regions with relatively lower temperature” The authors may rephrase this sentence as- Apples grown in lower temperature regions tend to have a high level of redness in apple skin however the same is not true in high temperature regions thereby reducing the cultivation and value of apples in these regions.

3.      Line 65 “…….. and a choice for the development of red fruit skin markers” not clear what authors wants to say here, rephrase the sentence.

4.      Label Figure 2c and d as well accordingly. Also, label the specific bands in Figure 2e to mark what they represent to conclude that hybrids with full bright red skin were Red-TE homozygous (RTRT) while hybrids with patched and medium-level red skin were Red-TE heterozygous (RTrt).

5.      Figure 3 a-h, move the scale bar to the right lower corner of the panel rather than placing it in the middle of 3 and f.

6.       There are spacing errors throughout the manuscript, check and make changes accordingly.

7.      The authors should read the manuscript thoroughly before resubmitting it, many sentences needs rephrasing throughout the manuscript, it’s not possible to point all of them out.

Reviewer 2 Report

  • In this study, the authors investigated the association between the presence of a Red-TE element on the promoter region of the MYB10 gene and the red colorization of apple skin. To support their hypothesis, the authors employed both resequencing and transcriptome approaches and identified putative alleles of MYB10 for each cultivar. Additionally, the developed primer set was able to effectively discriminate between homozygous and heterozygous Red-TE alleles and suggested a potential role for the TE in activating MYB10. However, it would be necessary to demonstrate the upregulation of MYB10 in conjunction with the presence of Red-TE across the tested cultivars. As the authors noted, the wild accession used in this study may have a different genome composition compared to the reference genome, which could potentially lead to incorrect results and the need to reconstruct the wild reference genome for further validation.
  • It would be beneficial to include layer information in Figure 1 to make it more informative. Additionally, the duplicated sequences in the genome browser are difficult to discern, and it would be helpful to include a flow chart to more clearly depict the deduction process.
  • It would be helpful to include the expression level of MYB10 in Figure 1e by utilizing additional transcriptome or qRT-PCR data.
  • It would be useful to include the genotypes of NY543, Royal Gala, and Granny Smith in Table 1.

Round 2

Reviewer 2 Report

The presented arguments were partially addressed, but the issue with sample collection can be understood.

There have been no objections to the publication of this paper.